# The Limits of Learning with Missing Data

**Brian Bullins**     **Elad Hazan**
Princeton University
Princeton, NJ
{bbullins,ehazan}@cs.princeton.edu

**Tomer Koren**
Google Brain
Mountain View, CA
tkoren@google.com

## Abstract

We study linear regression and classification in a setting where the learning algorithm is allowed to access only a limited number of attributes per example, known as the limited attribute observation model. In this well-studied model, we provide the first lower bounds giving a limit on the precision attainable by any algorithm for several variants of regression, notably linear regression with the absolute loss and the squared loss, as well as for classification with the hinge loss. We complement these lower bounds with a general purpose algorithm that gives an upper bound on the achievable precision limit in the setting of learning with missing data.

## 1 Introduction

The primary objective of linear regression is to determine the relationships between multiple variables and how they may affect a certain outcome. A standard example is that of medical diagnosis, whereby the data gathered for a given patient provides information about their susceptibility to certain illnesses. A major drawback to this process is the work necessary to collect the data, as it requires running numerous tests for each person, some of which may be discomforting. In such cases it may be necessary to impose limitations on the amount of data available for each example. For medical diagnosis, this might mean having each patient only undergo a small subset of tests.

A formal setting for capturing regression and learning with limits on the number of attribute observations is known as the Limited Attribute Observation (LAO) setting, first introduced by Ben-David and Dichterman [1]. For example, in a regression problem, the learner has access to a distribution $\mathcal{D}$ over data $(\mathbf{x}, y) \in \mathbb{R}^d \times \mathbb{R}$, and fits the best (generalized) linear model according to a certain loss function, i.e., it approximately solves the optimization problem

$$\min_{\mathbf{w}:\|\mathbf{w}\|_p \leq B} L_{\mathcal{D}}(\mathbf{w}), \qquad L_{\mathcal{D}}(\mathbf{w}) = \mathbb{E}_{(\mathbf{x},y)\sim\mathcal{D}} \left[ \ell(\mathbf{w}^\top \mathbf{x} - y) \right].$$

In the LAO setting, the learner does not have complete access to the examples $\mathbf{x}$, which the reader may think of as attributes of a certain patient. Rather, the learner can observe at most a fixed number of these attributes, denoted $k \leq d$. If $k = d$, this is the standard regression problem which can be solved to arbitrary precision.

The main question we address: is it possible to compute an arbitrarily accurate solution if the number of observations per example, $k$, is strictly less than $d$? More formally, given any $\varepsilon > 0$, can one compute a vector $\mathbf{w}$ for which

$$L_{\mathcal{D}}(\mathbf{w}) \leq \min_{\|\mathbf{w}^*\|_p \leq B} L_{\mathcal{D}}(\mathbf{w}^*) + \varepsilon.$$

Efficient algorithms for regression with squared loss when $k < d$ have been shown in previous work [2], and the sample complexity bounds have since been tightened [6, 8]. However, similar results for

other common loss functions such as e.g. absolute loss have only been shown by relaxing the hard limit of $k$ attributes per example [3, 6].

In this paper we show, for the first time, that in fact this problem cannot be solved in general. Our main result shows that even for regression with the absolute loss function, for any $k \leq d - 1$, there is an information-theoretic lower bound on the error attainable by any algorithm. That is, there is some $\varepsilon_0 > 0$ for which an $\varepsilon_0$-optimal solution cannot be determined, irrespective of the number of examples the learner sees. Formally, with constant probability, any algorithm returning a vector $\mathbf{w} \in \mathbb{R}^d$ must satisfy

$$L_{\mathcal{D}}(\mathbf{w}) > \min_{\|\mathbf{w}^*\|_p \leq B} L_{\mathcal{D}}(\mathbf{w}^*) + \varepsilon_0.$$

We further show that this ultimate achievable precision parameter is bounded from below by a polynomial in the dimension, i.e., $\varepsilon_0 = \Omega(d^{-3/2})$.

Additionally, for the basic setting of Ridge regression (with the squared loss), we give a tight lower bound for the LAO setting. Cesa-Bianchi et al. [2] provided the first efficient algorithm for this setting with sample complexity of $O(d^2/k\varepsilon^2)$ for $\varepsilon$ error. Hazan and Koren [6] improved upon this result and gave a tight sample complexity of $O(d/k\varepsilon^2)$ to achieve $\varepsilon$ error. In both cases, however, the algorithms only work when $k \geq 2$. We complete the picture and show that $k \geq 2$ attributes are in fact necessary to obtain arbitrarily low error. That is, with only one attribute per example, there is an information-theoretic limit on the accuracy attainable by *any* regression algorithm. We remark that a similar impossibility result was proven by Cesa-Bianchi et al. [3] in the related setting of learning with noisy examples.

Classification may be similarly cast in the LAO setting. For classification with the hinge loss, namely soft-margin SVM, we give a related lower bound, showing that it is impossible to achieve arbitrarily low error if the number of observed attributes is bounded by $k \leq d - 1$. However, unlike our lower bound for regression, the lower bound we prove for classification scales exponentially with the dimension. Although Hazan et al. [7] showed how classification may be done with missing data, their work includes low rank assumptions and so it is not in contradiction with the lower bounds presented here.

Similar to the LAO setting, the setting of learning with missing data [9, 4, 10, 11] presents the learner with examples where the attributes are randomly observed. Since the missing data setting is at least as difficult as the LAO setting, our lower bounds extend to this case as well.

We complement these lower bounds with a general purpose algorithm for regression and classification with missing data that, given a sufficient number of samples, can achieve an error of $O(1/\sqrt{d})$. This result leaves only a small polynomial gap compared to the information-theoretic lower bound that we prove.

## 2 Setup and Statement of Results

The general framework of linear regression involves a set of instances, each of the form $(\mathbf{x}, y)$ where $\mathbf{x} \in \mathbb{R}^d$ is the attribute vector and $y \in \mathbb{R}$ is the corresponding target value. Under the typical statistical learning framework [5], each $(\mathbf{x}, y)$ pair is drawn from a joint distribution $\mathcal{D}$ over $\mathbb{R}^d \times \mathbb{R}$. The learner's objective is to determine some linear predictor $\mathbf{w}$ such that $\mathbf{w}^\top \mathbf{x}$ does well in predicting $y$. The quality of prediction is measured according to a loss function $\ell : \mathbb{R} \mapsto \mathbb{R}$. Two commonly used loss functions for regression are the squared loss $\ell(\mathbf{w}^\top \mathbf{x} - y) = \frac{1}{2}(\mathbf{w}^\top \mathbf{x} - y)^2$ and the absolute loss $\ell(\mathbf{w}^\top \mathbf{x} - y) = |\mathbf{w}^\top \mathbf{x} - y|$. Since our examples are drawn from some arbitrary distribution $\mathcal{D}$, it is best to consider the expected loss

$$L_{\mathcal{D}}(\mathbf{w}) = \mathbb{E}_{(\mathbf{x}, y) \sim \mathcal{D}}\left[\ell(\mathbf{w}^\top \mathbf{x} - y)\right].$$

The learner's goal then is to determine a regressor $\mathbf{w}$ that minimizes the expected loss $L_{\mathcal{D}}(\mathbf{w})$. To avoid overfitting, a regularization term is typically added, which up to some constant factor is equivalent to

$$\min_{\mathbf{w} \in \mathbb{R}^d} L_{\mathcal{D}}(\mathbf{w}) \text{ s.t. } \|\mathbf{w}\|_p \leq B$$

for some regularization parameter $B > 0$, where $\| \cdot \|_p$ is the standard $\ell_p$ norm, $p \geq 1$. Two common variants of regression are Ridge regression ($p = 2$ with squared loss) and Lasso regression ($p = 1$ with squared loss).

The framework for classification is nearly identical to that of linear regression. The main distinction comes from a different meaning of $y \in \mathbb{R}$, namely that $y$ acts as a label for the corresponding example. The loss function also changes when learning a classifier, and in this paper we are interested in the hinge loss $\ell(y \cdot \mathbf{w}^\top \mathbf{x}) = \max\{0, 1 - y \cdot \mathbf{w}^\top \mathbf{x}\}$. The overall goal of the learner, however, remains the same: namely, to determine a classifier $\mathbf{w}$ such that $L_\mathcal{D}(\mathbf{w})$ is minimized. Throughout the paper, we let $\mathbf{w}^*$ denote the minimizer of $L_\mathcal{D}(\mathbf{w})$.

## 2.1 Main Results

As a first step, for Lasso and Ridge regressions, we show that one always needs to observe at least two attributes to be able to learn a regressor to arbitrary precision. This is given formally in Theorem 1.

**Theorem 1.** *Let $0 < \varepsilon < \frac{1}{32}$ and let $\ell$ be the squared loss. Then there exists a distribution $\mathcal{D}$ over $\{\mathbf{x} : ||\mathbf{x}||_\infty \leq 1\} \times [-1, 1]$ such that $||\mathbf{w}^*||_1 \leq 2$, and any regression algorithm that can observe at most one attribute of each training example of a training set $S$ cannot output a regressor $\hat{\mathbf{w}}$ such that $\mathbb{E}_S[L_\mathcal{D}(\hat{\mathbf{w}})] < L_\mathcal{D}(\mathbf{w}^*) + \varepsilon$.*

**Corollary 2.** *Let $0 < \varepsilon < \frac{1}{64}$ and let $\ell$ be the squared loss. Then there exists a distribution $\mathcal{D}$ over $\{\mathbf{x} : ||\mathbf{x}||_2 \leq 1\} \times [-1, 1]$ such that $||\mathbf{w}^*||_2 \leq 2$, and any regression algorithm that can observe at most one attribute of each training example of a training set $S$ cannot output a regressor $\hat{\mathbf{w}}$ such that $\mathbb{E}_S[L_\mathcal{D}(\hat{\mathbf{w}})] < L_\mathcal{D}(\mathbf{w}^*) + \varepsilon$.*

The lower bounds are tight—recall that with two attributes, it is indeed possible to learn a regressor to within arbitrary precision [2, 6]. Also, notice the order of quantification in the theorems: it turns out that there exists a distribution which is hard for all algorithms (rather than a different hard distribution for any algorithm).

For regression with absolute loss, we consider the setting where the learner is limited to seeing $k$ or fewer attributes of each training sample. Theorem 3 below shows that in the case where $k < d$ the learner cannot hope to learn an $\varepsilon$-optimal regressor for some $\varepsilon > 0$.

**Theorem 3.** *Let $d \geq 4$, $d \equiv 0 \pmod 2$, $0 < \varepsilon < \frac{1}{60} d^{-\frac{3}{2}}$, and let $\ell$ be the absolute loss. Then there exists a distribution $\mathcal{D}$ over $\{\mathbf{x} : ||\mathbf{x}||_\infty \leq 1\} \times [-1, 1]$ such that $||\mathbf{w}^*||_1 \leq 2$, and any regression algorithm that can observe at most $d - 1$ attributes of each training example of a training set $S$ cannot output a regressor $\hat{\mathbf{w}}$ such that $\mathbb{E}_S[L_\mathcal{D}(\hat{\mathbf{w}})] < L_\mathcal{D}(\mathbf{w}^*) + \varepsilon$.*

**Corollary 4.** *Let $0 < \varepsilon < \frac{1}{60} d^{-2}$, and let $\ell$ be the absolute loss. Then there exists a distribution $\mathcal{D}$ over $\{\mathbf{x} : ||\mathbf{x}||_2 \leq 1\} \times [-1, 1]$ such that $||\mathbf{w}^*||_2 \leq 1$, and any regression algorithm that can observe at most $d - 1$ attributes of each training example of a training set $S$ cannot output a regressor $\hat{\mathbf{w}}$ such that $\mathbb{E}_S[L_\mathcal{D}(\hat{\mathbf{w}})] < L_\mathcal{D}(\mathbf{w}^*) + \varepsilon$.*

We complement our findings for regression with the following analogous lower bound for classification with the hinge loss (a.k.a., soft margin SVM).

**Theorem 5.** *Let $d \geq 4$, $d \equiv 0 \pmod 2$, and let $\ell$ be the hinge loss. Then there exists an $\varepsilon_0 > 0$ such that the following holds: there exists a distribution $\mathcal{D}$ over $\{\mathbf{x} : ||\mathbf{x}||_2 \leq 1\} \times [-1, 1]$ such that $||\mathbf{w}^*||_2 \leq 1$, and any classification algorithm that can observe at most $d - 1$ attributes of each training example of a training set $S$ cannot output a regressor $\hat{\mathbf{w}}$ such that $\mathbb{E}_S[L_\mathcal{D}(\hat{\mathbf{w}})] < L_\mathcal{D}(\mathbf{w}^*) + \varepsilon_0$.*

## 3 Lower Bounds

In this section we discuss our lower bounds for regression with missing attributes. As a warm-up, we first prove Theorem 1 for regression with the squared loss. While the proof is very simple, it illustrates some of the main ideas used in all of our lower bounds. Then, we give a proof of Theorem 3 for regression with the absolute loss. The proofs of the remaining bounds are deferred to the supplementary material.

### 3.1 Lower bounds for the squared loss

*Proof of Theorem 1.* It is enough to prove the theorem for deterministic learning algorithms, namely, for algorithms that do not use any external randomization (i.e., any randomization besides the random samples drawn from the data distribution itself). This is because any randomized algorithm can

be thought of as a distribution over deterministic algorithms, which is independent of the data distribution.

Now, suppose $0 < \varepsilon < \frac{1}{32}$. Let $X_1 = \{(0,0),(1,1)\}$, $X_2 = \{(0,1),(1,0)\}$, and let $\mathcal{D}_1$ and $\mathcal{D}_2$ be uniform distributions over $X_1 \times \{1\}$ and $X_2 \times \{1\}$, respectively. The main observation is that any learner that can observe at most one attribute of each example cannot distinguish between the two distributions with probability greater than $\frac{1}{2}$, no matter how many samples it is given. This is because the marginal distributions of the individual attributes under both $\mathcal{D}_1$ and $\mathcal{D}_2$ are exactly the same. Thus, to prove the theorem it is enough to show that the sets of $\varepsilon$-optimal solutions under the distributions $\mathcal{D}_1$ and $\mathcal{D}_2$ are disjoint. Indeed, suppose that there is a learning algorithm that emits a vector $\hat{\mathbf{w}}$ such that $\mathbb{E}[L_{\mathcal{D}}(\hat{\mathbf{w}}) - L_{\mathcal{D}}(\mathbf{w}^*)] < \varepsilon/2$ (where the expectation is over the random samples from $\mathcal{D}$ used by the algorithm). By Markov's inequality, it holds that $L_{\mathcal{D}}(\hat{\mathbf{w}}) < L_{\mathcal{D}}(\mathbf{w}^*) + \varepsilon$ with probability $> 1/2$. Hence, the output of the algorithm allows one to distinguish between the two distributions with probability $> 1/2$, contradicting the indistinguishability property.

We set to characterize the sets of $\varepsilon$-optimal solutions under $\mathcal{D}_1$ and $\mathcal{D}_2$. For $\mathcal{D}_1$, we have

$$L_{\mathcal{D}_1}(\mathbf{w}) = \frac{1}{2} \sum_{\mathbf{x} \in X_1} \frac{1}{2} (\mathbf{w}^\top \mathbf{x} - 1)^2 = \frac{1}{4} + \frac{1}{4}(w_1 + w_2 - 1)^2,$$

while for $\mathcal{D}_2$,

$$L_{\mathcal{D}_2}(\mathbf{w}) = \frac{1}{2} \sum_{\mathbf{x} \in X_2} \frac{1}{2} (\mathbf{w}^\top \mathbf{x} - 1)^2 = \frac{1}{4}(w_1 - 1)^2 + \frac{1}{4}(w_2 - 1)^2.$$

Note that the set of $\varepsilon$-optimal regressors for $L_{\mathcal{D}_1}$ is $S_1 = \{\mathbf{w} : |\mathbf{w}^\top \mathbf{1} - 1| \leq 2\sqrt{\varepsilon}\}$, whereas for $L_{\mathcal{D}_2}$ the set is $S_2 = \{\mathbf{w} : \|\mathbf{w} - \mathbf{1}\|_2 \leq 2\sqrt{\varepsilon}\}$. Let $S_2' = \{\mathbf{w} : |\mathbf{w}^\top \mathbf{1} - 2| \leq 2\sqrt{2\varepsilon}\}$. Then $S_2 \subseteq S_2'$, so it is sufficient to show that $S_1$ and $S_2'$ are disjoint.

Since $\varepsilon < \frac{1}{32}$, for any $\mathbf{w} \in S_1$, $|\mathbf{w}^\top \mathbf{1} - 1| < \frac{1}{2}$, meaning $\mathbf{w}^\top \mathbf{1} < \frac{3}{2}$. However, for any $\mathbf{w} \in S_2'$, $|\mathbf{w}^\top \mathbf{1} - 2| < \frac{1}{2}$ meaning $\mathbf{w}^\top \mathbf{1} > \frac{3}{2}$, and so $\mathbf{w}$ cannot be a member of both $S_1$ and $S_2$. As we argued earlier, this suffices to prove the theorem. $\qquad\square$

## 3.2 Lower bounds for the absolute loss

As in the proof of Theorem 1, the main idea is to show that one can design two distributions that are indistinguishable to a learner who can observe no more than $d - 1$ attributes of any sample given by the distribution (i.e., that their marginals over any choice of $d - 1$ attributes are identical), but whose respective sets of $\varepsilon$-optimal regressors are disjoint. However, in contrast to Theorem 1, both handling general $d$ along with switching to the absolute loss introduce additional complexities to the proof that require different techniques.

We start by constructing these two distributions $\mathcal{D}_1$ and $\mathcal{D}_2$. Let $X_1 = \{\mathbf{x} = (x_1, \ldots, x_d) : \mathbf{x} \in \{0,1\}^d, \|\mathbf{x}\|_1 \equiv 0 \pmod 2\}$ and $X_2 = \{\mathbf{x} = (x_1, \ldots, x_d) : \mathbf{x} \in \{0,1\}^d, \|\mathbf{x}\|_1 \equiv 1 \pmod 2\}$, and let $\mathcal{D}_1$ and $\mathcal{D}_2$ be uniform over $X_1 \times \{1\}$ and $X_2 \times \{1\}$, respectively. From this construction, it is not hard to see that for any choice of $k \leq d - 1$ attributes, the marginals over the $k$ attributes of both distributions are identical: they are both a uniform distribution over $k$ bits. Thus, the distributions $\mathcal{D}_1$ and $\mathcal{D}_2$ are indistinguishable to a learner that can only observe at most $d - 1$ attributes of each example.

Let $\ell(\mathbf{w}^\top \mathbf{x} - y) = |\mathbf{w}^\top \mathbf{x} - y|$, and let

$$L_{\mathcal{D}_1}(\mathbf{w}) = \mathbb{E}_{(\mathbf{x},y) \sim \mathcal{D}_1}[\ell(\mathbf{w}^\top \mathbf{x}, y)] = \frac{1}{2^{d-1}} \sum_{\mathbf{x} \in X_1} |\mathbf{w}^\top \mathbf{x} - 1|,$$

and

$$L_{\mathcal{D}_2}(\mathbf{w}) = \mathbb{E}_{(\mathbf{x},y) \sim \mathcal{D}_2}[\ell(\mathbf{w}^\top \mathbf{x}, y)] = \frac{1}{2^{d-1}} \sum_{\mathbf{x} \in X_2} |\mathbf{w}^\top \mathbf{x} - 1|.$$

It turns out that the subgradients of $L_{\mathcal{D}_1}(\mathbf{w})$ and $L_{\mathcal{D}_2}(\mathbf{w})$, which we denote by $\partial L_{\mathcal{D}_1}(\mathbf{w})$ and $\partial L_{\mathcal{D}_2}(\mathbf{w})$ respectively, can be expressed precisely. In fact, the full subgradient set at every point in the domain for both functions can be made explicit. With these representations in hand, we can show that $\mathbf{w}_1^* = \frac{2}{d} \mathbf{1}_d$ and $\mathbf{w}_2^* = \frac{2}{d+2} \mathbf{1}_d$ are minimizers of $L_{\mathcal{D}_1}(\mathbf{w})$ and $L_{\mathcal{D}_2}(\mathbf{w})$, respectively.

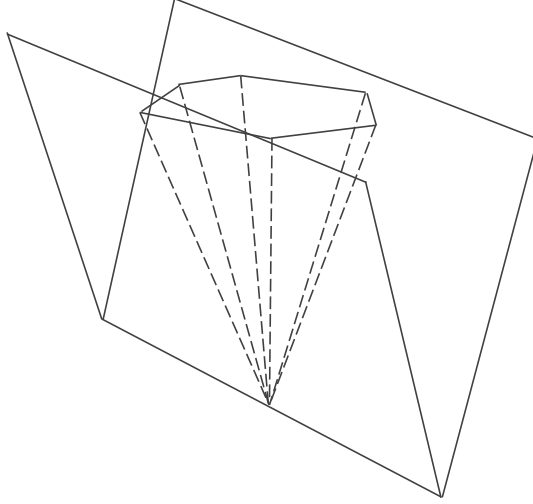

Figure 1: Geometric intuition for Lemmas 6 and 7. The lower bounding absolute value function acts as a relaxation of the true expected loss $L_\mathcal{D}$ (depicted here as a cone).

In fact, using the subgradient sets we can prove a much stronger property of the expected losses $L_{\mathcal{D}_1}$ and $L_{\mathcal{D}_2}$, akin to a "directional strong convexity" property around their respective minimizers. The geometric idea behind this property is shown in Figure 1, whereby $L_\mathcal{D}$ is lower bounded by an absolute value function.

**Lemma 6.** *Let* $\mathbf{w}_1^* = \frac{2}{d}\mathbf{1}_d$. *For any* $w \in \mathbb{R}^d$ *we have*

$$L_{\mathcal{D}_1}(\mathbf{w}) - L_{\mathcal{D}_1}(\mathbf{w}_1^*) \geq \frac{\sqrt{2\pi}}{e^4\sqrt{d}} \cdot \left|\mathbf{1}_d^\top(\mathbf{w} - \mathbf{w}_1^*)\right|.$$

**Lemma 7.** *Let* $\mathbf{w}_2^* = \frac{2}{d+2}\mathbf{1}_d$. *For any* $w \in \mathbb{R}^d$ *we have*

$$L_{\mathcal{D}_2}(\mathbf{w}) - L_{\mathcal{D}_2}(\mathbf{w}_2^*) \geq \frac{\sqrt{2\pi}}{e^4\sqrt{d}} \cdot \left|\mathbf{1}_d^\top(\mathbf{w} - \mathbf{w}_2^*)\right|.$$

Given Lemmas 6 and 7, the proof of Theorem 3 is immediate.

*Proof of Theorem 3.* As a direct consequence of Lemmas 6 and 7, we obtain that the sets

$$S_1 = \left\{\mathbf{w} \;:\; \left|\frac{\sqrt{2\pi}}{e^4\sqrt{d}} \cdot \mathbf{1}_d^\top(\mathbf{w} - \mathbf{w}_1^*)\right| \leq \varepsilon\right\}$$

and

$$S_2 = \left\{\mathbf{w} \;:\; \left|\frac{\sqrt{2\pi}}{e^4\sqrt{d}} \cdot \mathbf{1}_d^\top(\mathbf{w} - \mathbf{w}_2^*)\right| \leq \varepsilon\right\}$$

contain the sets of $\varepsilon$-optimal regressors for $L_{\mathcal{D}_1}(\mathbf{w})$ and $L_{\mathcal{D}_2}(\mathbf{w})$, respectively. All that is needed now is to show a separation of their $\varepsilon$-optimal sets for $0 < \varepsilon < \frac{1}{60}d^{-\frac{3}{2}}$, and this is done by showing a separation of the more manageable sets $S_1$ and $S_2$. Indeed, fix $0 < \varepsilon < \frac{1}{60}d^{-\frac{3}{2}}$ and observe that for any $\mathbf{w} \in S_1$ we have $-\frac{\sqrt{2\pi}}{e^4\sqrt{d}} \cdot \mathbf{1}_d^\top(\mathbf{w} - \mathbf{w}_1^*) \leq \frac{1}{60}d^{-\frac{3}{2}}$ and so, for $d \geq 4$,

$$\mathbf{1}_d^\top\mathbf{w} \geq 2 - \frac{1}{2d} > 2 - \frac{1}{d+2} = \frac{2d+3}{d+2}.$$

On the other hand, for any $\mathbf{w} \in S_2$ we have $\frac{\sqrt{2\pi}}{e^4\sqrt{d}} \cdot \mathbf{1}_d^\top(\mathbf{w} - \mathbf{w}_2^*) \leq \frac{1}{60}d^{-\frac{3}{2}}$, thus

$$\mathbf{1}_d^\top\mathbf{w} \leq \frac{2d}{d+2} + \frac{1}{2d} < \frac{2d}{d+2} + \frac{1}{d+2} = \frac{2d+1}{d+2}.$$

We see that no $\mathbf{w}$ can exist in both $S_1$ and $S_2$, so these sets are disjoint. Theorem 3 follows by the same reasoning used to conclude the proof of Theorem 1. □

It remains to prove Lemmas 6 and 7. As the proofs are very similar, we will only prove Lemma 6 here and defer the proof of Lemma 7 to the supplementary material.

*Proof of Lemma 6.* We first write

$$\partial L_{\mathcal{D}_1}(\mathbf{w}) = \frac{1}{2^{d-1}} \sum_{\mathbf{x} \in \mathcal{X}_1} \partial \ell(\mathbf{w}^\top \mathbf{x}, 1) = \frac{1}{2^{d-1}} \sum_{\mathbf{x} \in \mathcal{X}_1} \text{sign}(\mathbf{w}^\top \mathbf{x} - 1) \cdot \mathbf{x}.$$

Letting $\mathbf{w}_1^* = \frac{2}{d} \cdot \mathbf{1}_d$, we have that

$$\begin{aligned}
\partial L_{\mathcal{D}_1}(\mathbf{w}_1^*) &= \frac{1}{2^{d-1}} \sum_{\mathbf{x} \in \mathcal{X}_1} \text{sign}(\mathbf{w}_1^{*\top} \mathbf{x} - 1) \cdot \mathbf{x} \\
&= \frac{1}{2^{d-1}} \bigg( \sum_{\substack{\mathbf{x} \in \mathcal{X}_1, \\ \|\mathbf{x}\|_1 = \frac{d}{2}}} \text{sign}(\mathbf{w}_1^{*\top} \mathbf{x} - 1) \cdot \mathbf{x} \\
&\qquad + \sum_{\substack{\mathbf{x} \in \mathcal{X}_1, \\ \|\mathbf{x}\|_1 > \frac{d}{2}}} \text{sign}(\mathbf{w}_1^{*\top} \mathbf{x} - 1) \cdot \mathbf{x} + \sum_{\substack{\mathbf{x} \in \mathcal{X}_1, \\ \|\mathbf{x}\|_1 < \frac{d}{2}}} \text{sign}(\mathbf{w}_1^{*\top} \mathbf{x} - 1) \cdot \mathbf{x} \bigg) \\
&= \frac{1}{2^{d-1}} \bigg( \sum_{\substack{\mathbf{x} \in \mathcal{X}_1, \\ \|\mathbf{x}\|_1 = \frac{d}{2}}} \text{sign}(0) \cdot \mathbf{x} + \sum_{\substack{\mathbf{x} \in \mathcal{X}_1, \\ \|\mathbf{x}\|_1 > \frac{d}{2}}} \mathbf{x} - \sum_{\substack{\mathbf{x} \in \mathcal{X}_1, \\ \|\mathbf{x}\|_1 < \frac{d}{2}}} \mathbf{x} \bigg),
\end{aligned}$$

where $\text{sign}(0)$ can be any number in $[-1, 1]$. Next, we compute

$$\begin{aligned}
\sum_{\substack{\mathbf{x} \in \mathcal{X}_1, \\ \|\mathbf{x}\|_1 > \frac{d}{2}}} \mathbf{x} - \sum_{\substack{\mathbf{x} \in \mathcal{X}_1, \\ \|\mathbf{x}\|_1 < \frac{d}{2}}} \mathbf{x} &= \sum_{i=\frac{d}{4}+1}^{\frac{d}{2}} \binom{d-1}{2i-1} \cdot \mathbf{1}_d - \sum_{i=1}^{\frac{d}{4}-1} \binom{d-1}{2i-1} \cdot \mathbf{1}_d \\
&= \sum_{i=0}^{\frac{d}{2}-2} (-1)^i \binom{d-1}{i} \cdot \mathbf{1}_d \\
&= \binom{d-2}{\frac{d}{2}-2} \cdot \mathbf{1}_d,
\end{aligned}$$

where the last equality follows from the elementary identity $\sum_{i=0}^{k}(-1)^i \binom{n}{i} = (-1)^k \binom{n-1}{k}$, which we prove in Lemma 9 in the supplementary material. Now, let $\mathcal{X}^* = \{\mathbf{x} \in \mathcal{X}_1 : \|\mathbf{x}\|_1 = \frac{d}{2}\}$, let $m = |\mathcal{X}^*|$, and let $X = [\mathbf{x}_1, \ldots, \mathbf{x}_m] \in \mathbb{R}^{d \times m}$ be the matrix formed by all $\mathbf{x} \in \mathcal{X}^*$. Then we may express the entire subgradient set explicitly as

$$\partial L_{\mathcal{D}_1}(\mathbf{w}_1^*) = \left\{ \frac{1}{2^{d-1}} \left( X\mathbf{r} + \binom{d-2}{\frac{d}{2}-2} \cdot \mathbf{1}_d \right) \ \Big| \ \mathbf{r} \in [-1, 1]^m \right\}.$$

Thus, any choice of $\mathbf{r} \in [-1, 1]^m$ will result in a specific subgradient of $L_{\mathcal{D}_1}(\mathbf{w}_1^*)$. Consider two such choices: $\mathbf{r}_1 = \mathbf{0}$ and $\mathbf{r}_2 = -\mathbf{1}_d$. Note that $X\mathbf{r}_1 = \mathbf{0}$ and $X\mathbf{r}_2 = -\binom{d-1}{\frac{d}{2}-1} \cdot \mathbf{1}_d$; to see the last equality, consider any fixed coordinate $i$ and notice that the number of elements in $\mathcal{X}^*$ with non-zero values in the $i$'th coordinate is equal to the number of ways to choose the remaining $\frac{d}{2} - 1$ non-zero coordinates from the other $d - 1$ coordinates. We then observe that the corresponding subgradients are

$$\mathbf{h}^+ = \frac{1}{2^{d-1}} \left( X\mathbf{r}_1 + \binom{d-2}{\frac{d}{2}-2} \cdot \mathbf{1}_d \right) = \frac{1}{2^{d-1}} \binom{d-2}{\frac{d}{2}-2} \cdot \mathbf{1}_d,$$

and

$$\mathbf{h}^- = \frac{1}{2^{d-1}} \left( X\mathbf{r}_2 + \binom{d-2}{\frac{d}{2}-2} \cdot \mathbf{1}_d \right) = -\frac{1}{2^{d-1}} \binom{d-2}{\frac{d}{2}-1} \cdot \mathbf{1}_d.$$

Note that, since the set of subgradients of $L_{\mathcal{D}_1}(\mathbf{w}_1^*)$ is a convex set, by taking a convex combination of $\mathbf{h}^+$ and $\mathbf{h}^-$ it follows that $\mathbf{0} \in \partial L_{\mathcal{D}_1}(\mathbf{w}_1^*)$ and so we see that $\mathbf{w}_1^*$ is a minimizer of $L_{\mathcal{D}_1}(\mathbf{w})$.

Given a handle on the subgradient set, we now show that these coefficients are polynomial in $d$. Observe that, using the fact that $\sqrt{2\pi n}(\frac{n}{e})^n \le n! \le e\sqrt{n}(\frac{n}{e})^n$, we have

$$\frac{1}{2^{d-1}}\binom{d-2}{\frac{d}{2}-2} \ge \frac{1}{2^{d-1}}\left(\frac{\sqrt{2\pi(d-2)}\left(\frac{d-2}{e}\right)^{d-2}}{e^2\sqrt{\frac{d-4}{2}}\sqrt{\frac{d}{2}}\left(\frac{d-4}{2e}\right)^{\frac{d-4}{2}-2}\left(\frac{d}{2e}\right)^{\frac{d}{2}}}\right)$$

$$\ge \frac{1}{2^{d-1}}\left(\frac{\sqrt{2\pi}}{e^2\sqrt{d}\left(\frac{1}{2^{d-1}}\right)}\right)\left(\frac{d-2}{d}\right)^{d-2}$$

$$\ge \left(\frac{\sqrt{2\pi}}{e^2\sqrt{d}}\right)\left(1-\frac{2}{d-2}\right)^{d-2}$$

$$\ge \frac{\sqrt{2\pi}}{e^4\sqrt{d}}.$$

Let $\mathbf{h}^* = \frac{\sqrt{2\pi}}{e^4\sqrt{d}}\cdot\mathbf{1}_d$. Since $\mathbf{h}^*$ can be written as a convex combination of $\mathbf{h}^+$ and $\mathbf{0}$, we see that $\mathbf{h}^* \in \partial L_{\mathcal{D}_1}(\mathbf{w}_1^*)$. Similarly we may see that

$$-\frac{1}{2^{d-1}}\binom{d-2}{\frac{d}{2}-1} \le -\frac{1}{2^{d-1}}\left(\frac{\sqrt{2\pi(d-2)}\left(\frac{d-2}{e}\right)^{d-2}}{e^2(\frac{d}{2}-1)\left(\frac{d-2}{2e}\right)^{d-2}}\right) = -\frac{\sqrt{2\pi}}{e^2\sqrt{d-2}} \le -\frac{\sqrt{2\pi}}{e^4\sqrt{d}}.$$

Again, since $-\mathbf{h}^*$ can be written as a convex combination of the vectors $\mathbf{h}^-$ and $\mathbf{0}$ in the subgradient set, we may conclude that $-\mathbf{h}^* \in \partial L_{\mathcal{D}_1}(\mathbf{w}_1^*)$ as well.

By the subgradient inequality it follows that, for all $\mathbf{w} \in \mathbb{R}^d$,

$$L_{\mathcal{D}_1}(\mathbf{w}) - L_{\mathcal{D}_1}(\mathbf{w}_1^*) \ge \mathbf{h}^{*\top}(\mathbf{w}-\mathbf{w}_1^*) = \frac{\sqrt{2\pi}}{e^4\sqrt{d}}\cdot\mathbf{1}_d^\top(\mathbf{w}-\mathbf{w}_1^*)$$

and

$$L_{\mathcal{D}_1}(\mathbf{w}) - L_{\mathcal{D}_1}(\mathbf{w}_1^*) \ge -\mathbf{h}^{*\top}(\mathbf{w}-\mathbf{w}_1^*) = -\frac{\sqrt{2\pi}}{e^4\sqrt{d}}\cdot\mathbf{1}_d^\top(\mathbf{w}-\mathbf{w}_1^*),$$

which taken together imply that

$$L_{\mathcal{D}_1}(\mathbf{w}) - L_{\mathcal{D}_1}(\mathbf{w}_1^*) \ge \frac{\sqrt{2\pi}}{e^4\sqrt{d}}\cdot\left|\mathbf{1}_d^\top(\mathbf{w}-\mathbf{w}_1^*)\right|$$

as required. □

## 4  General Algorithm for Limited Precision

Although we have established limits on the attainable precision for some learning problems, there is still the possibility of reaching this limit. In this section we provide a general algorithm, whereby a learner that can observe $k < d$ attributes can always achieve an expected loss of $O(\sqrt{1-k/d})$.

We provide the pseudo-code in Algorithm 1. Although similar to the AERR algorithm of Hazan and Koren [6]—which is designed to work only with the squared loss—Algorithm 1 avoids the necessity of an unbiased gradient estimator by replacing the original loss function with a slightly biased one. As long as the new loss function is chosen carefully (and the functions are Lipschitz bounded), and given enough samples, the algorithm can return a regressor of limited precision. This is in contrast to AERR whereby an arbitrarily precise regressor can always be achieved with enough samples.

Formally, for Algorithm 1 we prove the following (proof in the supplementary material).

**Theorem 8.** *Let $\ell : \mathbb{R} \mapsto \mathbb{R}$ be an $H$-Lipschitz function defined over $[-2B, 2B]$. Assume the distribution $\mathcal{D}$ is such that $\|\mathbf{x}\|_2 \le 1$ and $|y| \le B$ with probability 1. Let $\tilde{B} = \max\{B, 1\}$, and let $\hat{\mathbf{w}}$ be the output of Algorithm 1, when run with $\eta = \frac{2B}{G\sqrt{m}}$. Then, $\|\hat{\mathbf{w}}\|_2 \le B$, and for any $\mathbf{w}^* \in \mathbb{R}^d$ with $\|\mathbf{w}^*\|_2 \le B$,*

$$\mathbb{E}[L_{\mathcal{D}}(\hat{\mathbf{w}})] \le L_{\mathcal{D}}(\mathbf{w}^*) + \frac{2HB}{\sqrt{m}} + 2H\tilde{B}^2\sqrt{1-\frac{k}{d}}.$$

**Algorithm 1** General algorithm for regression/classification with missing attributes

---

**Input:** Loss function $\ell$, training set $S = \{(\mathbf{x}_t, y_t)\}_{t \in [m]}$, $k, B, \eta > 0$
**Output:** Regressor $\hat{\mathbf{w}}$ with $\|\hat{\mathbf{w}}\|_2 \leq B$
 1: Initialize $\mathbf{w}_1 \neq \mathbf{0}$, $\|\mathbf{w}_1\|_2 \leq B$ arbitrarily
 2: **for** $t = 1$ to $m$ **do**
 3:     Uniformly choose subset of $k$ indices $\{i_{t,r}\}_{r \in [k]}$ from $[d]$ without replacement
 4:     Set $\tilde{\mathbf{x}}_t = \sum_{r=1}^{k} \mathbf{x}[i_{t,r}] \cdot \mathbf{e}_{i_{t,r}}$
 5:     **Regression case:**
 6:         Choose $\hat{\phi}_t \in \partial \ell(\mathbf{w}_t^\top \tilde{\mathbf{x}}_t - y_t)$
 7:     **Classification case:**
 8:         Choose $\hat{\phi}_t \in \partial \ell(y_t \cdot \mathbf{w}_t^\top \tilde{\mathbf{x}}_t)$
 9:     Update

$$\mathbf{w}_{t+1} = \frac{B}{\max\{\|\mathbf{w}_t - \eta(\hat{\phi}_t \cdot \tilde{\mathbf{x}}_t)\|_2, B\}} \cdot (\mathbf{w}_t - \eta(\hat{\phi}_t \cdot \tilde{\mathbf{x}}_t))$$

10: **end for**
11: Return $\hat{\mathbf{w}} = \frac{1}{m} \sum_{t=1}^{m} \mathbf{w}_t$

---

In particular, for $m = d/(d-k)$ we have

$$\mathbb{E}[L_{\mathcal{D}}(\hat{\mathbf{w}})] \leq L_{\mathcal{D}}(\mathbf{w}^*) + 4H\tilde{B}^2 \sqrt{1 - \frac{k}{d}},$$

and so when the learner observes $k = d - 1$ attributes, the expected loss is $O(1/\sqrt{d})$-away from optimum.

## 5   Conclusions and Future Work

In the limited attribute observation setting, we have shown information-theoretic lower bounds for some variants of regression, proving that a distribution-independent algorithm for regression with absolute loss that attains $\varepsilon$ error cannot exist and closing the gap for ridge regression as suggested by Hazan and Koren [6]. We have also shown that the proof technique applied for regression with absolute loss can be extended to show a similar bound for classification with the hinge loss. In addition, we have described a general purpose algorithm which complements these results by providing a means of achieving error up to a certain precision limit.

An interesting possibility for future work would be to try to bridge the gap between the upper and lower bounds of the precision limits, particularly in the case of the exponential gap for classification with hinge loss. Another direction would be to develop a more comprehensive understanding of these lower bounds in terms of more general functions, one example being classification with logistic loss.

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
