[Supplementary Material]

# The Limits of Learning with Missing Data (Supplement)

**Brian Bullins      Elad Hazan**
Princeton University
Princeton, NJ
{bbullins,ehazan}@cs.princeton.edu

**Tomer Koren**
Google Brain
Mountain View, CA
tkoren@google.com

## A    Proofs

### A.1    Proof of Theorem 3 (Remaining)

**Lemma 9.** *For any integers $0 \le k < d$ it holds that*

$$\sum_{i=0}^{k}(-1)^{i}\binom{d}{i} = (-1)^{k}\binom{d-1}{k}.$$

*Proof.* By induction. Clearly the $k = 0$ case holds. Now, suppose for some $0 \le k < d - 1$,

$$\sum_{i=0}^{k}(-1)^{i}\binom{d}{i} = (-1)^{k}\binom{d-1}{k}.$$

Then, we have

$$\sum_{i=0}^{k+1}(-1)^{i}\binom{d}{i} = (-1)^{k}\binom{d-1}{k} + (-1)^{k+1}\binom{d}{k+1} = (-1)^{k+1}\left(\binom{d}{k+1} - \binom{d-1}{k}\right) = (-1)^{k+1}\binom{d-1}{k+1},$$

as desired.                                                                                                       □

**Proof of Lemma 7**

*Proof.* The $\mathcal{D}_2$ case uses a line of reasoning similar to that we used for $\mathcal{D}_1$. Note that

$$\partial L_{\mathcal{D}_2}(\mathbf{w}) = \frac{1}{2^{d-1}} \sum_{\mathbf{x} \in \mathcal{X}_2} \partial \ell(\mathbf{w}^{\top}\mathbf{x} - 1) = \frac{1}{2^{d-1}} \sum_{\mathbf{x} \in \mathcal{X}_2} \mathrm{sign}(\mathbf{w}^{\top}\mathbf{x} - 1) \cdot \mathbf{x}$$

Now, letting $\mathbf{w}_2^* = \frac{2}{d+2} \cdot \mathbf{1}_d$, we have that

$$
\partial L_{\mathcal{D}_2}(\mathbf{w}_2^*) = \frac{1}{2^{d-1}} \sum_{\mathbf{x} \in \mathcal{X}_2} \partial \ell(\mathbf{w}_2^{*\top}\mathbf{x} - 1) = \frac{1}{2^{d-1}} \sum_{\mathbf{x} \in \mathcal{X}_2} \operatorname{sign}(\mathbf{w}_2^{*\top}\mathbf{x} - 1) \cdot \mathbf{x}
$$

$$
= \frac{1}{2^{d-1}} \Bigg( \sum_{\substack{\mathbf{x} \in \mathcal{X}_2, \\ \|\mathbf{x}\|_1 < \frac{d+2}{2}}} \operatorname{sign}(\mathbf{w}_2^{*\top}\mathbf{x} - 1) \cdot \mathbf{x}
$$

$$
+ \sum_{\substack{\mathbf{x} \in \mathcal{X}_2, \\ \|\mathbf{x}\|_1 = \frac{d+2}{2}}} \operatorname{sign}(\mathbf{w}_2^{*\top}\mathbf{x} - 1) \cdot \mathbf{x} + \sum_{\substack{\mathbf{x} \in \mathcal{X}_2, \\ \|\mathbf{x}\|_1 > \frac{d+2}{2}}} \operatorname{sign}(\mathbf{w}_2^{*\top}\mathbf{x} - 1) \cdot \mathbf{x} \Bigg)
$$

$$
= \frac{1}{2^{d-1}} \Bigg( \sum_{\substack{\mathbf{x} \in \mathcal{X}_2, \\ \|\mathbf{x}\|_1 = \frac{d+2}{2}}} \operatorname{sign}(0) \cdot \mathbf{x} + \sum_{\substack{\mathbf{x} \in \mathcal{X}_2, \\ \|\mathbf{x}\|_1 > \frac{d+2}{2}}} \mathbf{x} - \sum_{\substack{\mathbf{x} \in \mathcal{X}_2, \\ \|\mathbf{x}\|_1 < \frac{d+2}{2}}} \mathbf{x} \Bigg)
$$

$$
= \frac{1}{2^{d-1}} \Bigg( \sum_{\substack{\mathbf{x} \in \mathcal{X}_2, \\ \|\mathbf{x}\|_1 = \frac{d+2}{2}}} \operatorname{sign}(0) \cdot \mathbf{x} + \sum_{i=\frac{d}{4}+1}^{\frac{d}{2}-1} \binom{d-1}{2i} \cdot \mathbf{1}_d - \sum_{i=0}^{\frac{d}{4}-1} \binom{d-1}{2i} \cdot \mathbf{1}_d \Bigg)
$$

$$
= \frac{1}{2^{d-1}} \Bigg( \sum_{\substack{\mathbf{x} \in \mathcal{X}_2, \\ \|\mathbf{x}\|_1 = \frac{d+2}{2}}} \operatorname{sign}(0) \cdot \mathbf{x} + \sum_{i=0}^{\frac{d}{2}-2} (-1)^{i+1} \binom{d-1}{i} \cdot \mathbf{1}_d \Bigg)
$$

$$
= \frac{1}{2^{d-1}} \Bigg( \sum_{\substack{\mathbf{x} \in \mathcal{X}_2, \\ \|\mathbf{x}\|_1 = \frac{d+2}{2}}} \operatorname{sign}(0) \cdot \mathbf{x} - \binom{d-2}{\frac{d}{2}-2} \cdot \mathbf{1}_d \Bigg) .
$$

Similar to before, we have

$$
\partial L_{\mathcal{D}_2}(\mathbf{w}_2^*) = \left\{ \frac{1}{2^{d-1}} \left( X\mathbf{r} - \binom{d-2}{\frac{d}{2}-2} \cdot \mathbf{1}_d \right) \mid \mathbf{r} \in [-1, 1]^m \right\}
$$

and so choosing $\mathbf{r}_1 = \mathbf{0}$ and $\mathbf{r}_2 = \mathbf{1}_d$ and following the same reasoning as in Lemma 6, we have that both of the vectors

$$
\mathbf{h}^+ = \frac{1}{2^{d-1}} \binom{d-2}{\frac{d}{2}-1} \cdot \mathbf{1}_d \quad , \quad \mathbf{h}^- = -\frac{1}{2^{d-1}} \binom{d-2}{\frac{d}{2}-2} \cdot \mathbf{1}_d
$$

belong to the subdifferential set $\partial L_{\mathcal{D}_2}(\mathbf{w}_2^*)$, and thus so does the zero vector $\mathbf{0}$, that lies on the segment connecting $\mathbf{h}^+$ and $\mathbf{h}^-$. Again, using $\sqrt{2\pi n}(\frac{n}{e})^n \le n! \le e\sqrt{n}(\frac{n}{e})^n$, we can show that

$$
\frac{1}{2^{d-1}} \binom{d-2}{\frac{d}{2}-1} \ge \frac{1}{2^{d-1}} \left( \frac{\sqrt{2\pi(d-2)} \left(\frac{d-2}{e}\right)^{d-2}}{e^2(\frac{d}{2}-1) \left(\frac{d-2}{2e}\right)^{d-2}} \right) = \frac{\sqrt{2\pi}}{e^2 \sqrt{d-2}} \ge \frac{\sqrt{2\pi}}{e^4 \sqrt{d}},
$$

and

$$
-\frac{1}{2^{d-1}} \binom{d-2}{\frac{d}{2}-2} \le -\frac{1}{2^{d-1}} \left( \frac{\sqrt{2\pi(d-2)} \left(\frac{d-2}{e}\right)^{d-2}}{e^2 \sqrt{\frac{d-4}{2}} \sqrt{\frac{d}{2}} \left(\frac{d-4}{2e}\right)^{\frac{d}{2}-2} \left(\frac{d}{2e}\right)^{\frac{d}{2}}} \right) \le -\frac{\sqrt{2\pi}}{e^4 \sqrt{d}}.
$$

Letting $\mathbf{h}^* = -\frac{\sqrt{2\pi}}{e^4 \sqrt{d}} \cdot \mathbf{1}_d$, we again have that $\pm \mathbf{h}^* \in \partial L_{\mathcal{D}_2}(\mathbf{w}_2^*)$. From this point, it should be clear that the analysis for the $\mathcal{D}_2$ case is identical to that of the $\mathcal{D}_1$ case, and so the lemma holds for $\mathcal{D}_2$ as well. □

## A.2 Proof of Theorem 5

Similar to the proof of Theorem 3, the main idea is to construct two distributions that are indistinguishable to a learner who can observe no more than $d - 1$ attributes of any sample, but whose respective sets of $\varepsilon$-optimal classifiers are disjoint.

To prove the lower bound, we again consider two distributions $\mathcal{D}_1$ and $\mathcal{D}_2$. Let $\mathcal{X}_1 = \{\mathbf{x} = (x_1, \ldots, x_d) \mid \mathbf{x} \in \{0,1\}^d, \|\mathbf{x}\|_1 \equiv 0 \pmod 2\}$ and $\mathcal{X}_2 = \{\mathbf{x} = (x_1, \ldots, x_d) \mid \mathbf{x} \in \{0,1\}^d, \|\mathbf{x}\|_1 \equiv 1 \pmod 2\}$. We also let $\mathcal{X}_1' = \frac{1}{\sqrt{d}}\mathcal{X}_1$ and $\mathcal{X}_2' = \frac{1}{\sqrt{d}}\mathcal{X}_2$. Furthermore, let $\mathcal{Y}_1 = \mathcal{Y}_2 = \{-\frac{1}{\sqrt{d}}, \frac{1}{\sqrt{d}}\}$. Let $(\mathbf{x}, y) \sim \mathcal{D}_1$ be such that $\mathbf{x}$ and $y$ are sampled independently, and such that $\mathbf{x}$ is sampled uniformly from $\mathcal{X}_1'$ while $y = \frac{1}{\sqrt{d}}$ w.p. $\frac{1}{2} + \delta$ and $y = -\frac{1}{\sqrt{d}}$ w.p. $\frac{1}{2} - \delta$, where $\delta = \frac{1}{2^{d+1}}$. We construct the distribution $\mathcal{D}_2$ in a similar manner.

Again, we observe that for any choice of $k$ attributes, $k \leq d - 1$, the marginal distributions are identical in both $\mathcal{D}_1$ and $\mathcal{D}_2$. Let $\ell(y_t \cdot \mathbf{w}^\top \mathbf{x}) = \max\{0, 1 - y \cdot \mathbf{w}^\top \mathbf{x}\}$, and let

$$L_{\mathcal{D}_1}(\mathbf{w}) = \frac{1}{2^{d-1}\sqrt{d}} \sum_{\mathbf{x} \in \mathcal{X}_1} \left(\frac{1}{2} + \delta\right) \max\{0, 1 - \mathbf{w}^\top \mathbf{x}\} + \left(\frac{1}{2} - \delta\right) \max\{0, 1 + \mathbf{w}^\top \mathbf{x}\}$$

and

$$L_{\mathcal{D}_2}(\mathbf{w}) = \frac{1}{2^{d-1}\sqrt{d}} \sum_{\mathbf{x} \in \mathcal{X}_2} \left(\frac{1}{2} + \delta\right) \max\{0, 1 - \mathbf{w}^\top \mathbf{x}\} + \left(\frac{1}{2} - \delta\right) \max\{0, 1 + \mathbf{w}^\top \mathbf{x}\}.$$

The goal is to show that the sets of $\varepsilon$-optimal classifiers for each of these minimization problems are disjoint for small enough $\varepsilon$. We begin by characterizing supersets of these $\varepsilon$-optimal sets which are easier to compare. Once established, we then show that these sets are disjoint.

**Lemma 10.** *Let* $\mathbf{w}_1^* = \frac{1}{d} \cdot \mathbf{1}_d$, *and let* $S_1 = \left\{\mathbf{w} : \left|\frac{1}{2^{d+1}\sqrt{d}} \cdot \mathbf{1}_d^\top(\mathbf{w} - \mathbf{w}_1^*)\right| \leq \varepsilon\right\}$. *Then the entire set of $\varepsilon$-optimal classifiers for $L_{\mathcal{D}_1}$ is contained in $S_1$.*

*Proof.* Notice that since $\max\{x, y\} = \frac{1}{2}(x + y + |y - x|)$, we can rewrite $L_{\mathcal{D}_1}$ as

$$L_{\mathcal{D}_1}(\mathbf{w}) = \frac{1}{2^d\sqrt{d}} \sum_{\mathbf{x} \in \mathcal{X}_1} \left(\frac{1}{2} + \delta\right)(1 - \mathbf{w}^\top \mathbf{x} + |1 - \mathbf{w}^\top \mathbf{x}|) + \left(\frac{1}{2} - \delta\right)(1 + \mathbf{w}^\top \mathbf{x} + |1 + \mathbf{w}^\top \mathbf{x}|)$$

Note that

$$\partial L_{\mathcal{D}_1}(\mathbf{w}) = \frac{1}{2^d} \sum_{\mathbf{x} \in \mathcal{X}_1} \left(\frac{1}{2} + \delta\right)(-\mathbf{x} - \text{sign}(1 - \mathbf{w}^\top \mathbf{x}) \cdot \mathbf{x}) + \left(\frac{1}{2} - \delta\right)(\mathbf{x} + \text{sign}(1 + \mathbf{w}^\top \mathbf{x}) \cdot \mathbf{x})$$

$$= \frac{1}{2^d\sqrt{d}} \sum_{\mathbf{x} \in \mathcal{X}_1} -2\delta\mathbf{x} - \left(\frac{1}{2} + \delta\right)\text{sign}(1 - \mathbf{w}^\top \mathbf{x}) \cdot \mathbf{x} + \left(\frac{1}{2} - \delta\right)\text{sign}(1 + \mathbf{w}^\top \mathbf{x}) \cdot \mathbf{x}$$

$$= \frac{1}{\sqrt{d}}\left(-\frac{\delta}{2}\mathbf{1}_d - \frac{1}{2^d} \sum_{\mathbf{x} \in \mathcal{X}_1} \left(\frac{1}{2} + \delta\right)\text{sign}(1 - \mathbf{w}^\top \mathbf{x}) \cdot \mathbf{x} + \left(\frac{1}{2} - \delta\right)\text{sign}(1 + \mathbf{w}^\top \mathbf{x}) \cdot \mathbf{x}\right)$$

Let $\mathbf{x}^* = \mathbf{1}_d$, and let $\mathbf{w}_1^* = \frac{1}{d}\mathbf{1}_d$. Note that $\mathbf{x}^* \in \mathcal{X}_1$. Taken together, we see that

$$\partial L_{\mathcal{D}_1}(\mathbf{w}_1^*) = \frac{1}{\sqrt{d}}\left(-\frac{\delta}{2}\mathbf{1}_d - \frac{1}{2^d}\left(\frac{1}{2} + \delta\right)\text{sign}(1 - \frac{1}{d}\mathbf{1}_d^\top\mathbf{x}^*) \cdot \mathbf{x}^*\right.$$

$$\left. - \frac{1}{2^d} \sum_{\mathbf{x} \in \mathcal{X}_1 \setminus \mathbf{x}^*} \left(\frac{1}{2} + \delta\right)\text{sign}(1 - \frac{1}{d}\mathbf{1}_d^\top\mathbf{x}) \cdot \mathbf{x} + \frac{1}{2^d} \sum_{\mathbf{x} \in \mathcal{X}_1} \left(\frac{1}{2} - \delta\right)\text{sign}(1 + \frac{1}{d}\mathbf{1}_d^\top\mathbf{x}) \cdot \mathbf{x}\right)$$

$$= \frac{1}{\sqrt{d}}\left(-\frac{\delta}{2}\mathbf{1}_d - \frac{1}{2^d}\left(\frac{1}{2} + \delta\right)\text{sign}(0) \cdot \mathbf{1}_d - \left(\frac{1}{4} - \frac{1}{2^d}\right)\left(\frac{1}{2} + \delta\right) \cdot \mathbf{1}_d + \frac{1}{4}\left(\frac{1}{2} - \delta\right) \cdot \mathbf{1}_d\right)$$

$$= \frac{1}{\sqrt{d}}\left(-\delta + (1 - \text{sign}(0))\left(\frac{1 + 2\delta}{2^{d+1}}\right)\right) \cdot \mathbf{1}_d$$

Since $\delta = \frac{1}{2^{d+1}}$, and choosing $1 \in \text{sign}(0)$, we have that

$$\mathbf{h}^- = -\frac{1}{2^{d+1}\sqrt{d}} \cdot \mathbf{1}_d \in \partial L_{\mathcal{D}_1}(\mathbf{w}_1^*).$$

On the other hand, choosing $-1 \in \text{sign}(0)$, we have

$$\mathbf{h}^+ = \frac{1}{\sqrt{d}}\left(\frac{1}{2^{d+1}} + \frac{1}{4^d}\right) \cdot \mathbf{1}_d \in \partial L_{\mathcal{D}_1}(\mathbf{w}_1^*).$$

Since $\mathbf{0}_d$ can be written as a convex combination of $\mathbf{h}^-$ and $\mathbf{h}^+$, we know that $\mathbf{w}_1^*$ is a minimizer of $L_{\mathcal{D}_1}(\mathbf{w})$. By the subgradient inequality, we know that for all $\mathbf{y} \in \mathbf{dom}L_{\mathcal{D}_1}$,

$$L_{\mathcal{D}_1}(\mathbf{y}) - L_{\mathcal{D}_1}(\mathbf{w}_1^*) \geq \frac{1}{2^{d+1}\sqrt{d}} \cdot \mathbf{1}_d^\top(\mathbf{y} - \mathbf{w}_1^*)$$

and

$$L_{\mathcal{D}_1}(\mathbf{y}) - L_{\mathcal{D}_1}(\mathbf{w}_1^*) \geq -\frac{1}{2^{d+1}\sqrt{d}} \cdot \mathbf{1}_d^\top(\mathbf{y} - \mathbf{w}_1^*)$$

which taken together implies that

$$L_{\mathcal{D}_1}(\mathbf{y}) - L_{\mathcal{D}_1}(\mathbf{w}_1^*) \geq \left| \frac{1}{2^{d+1}\sqrt{d}} \cdot \mathbf{1}_d^\top(\mathbf{y} - \mathbf{w}_1^*) \right|.$$

Now, letting $S_1 = \left\{ \mathbf{w} \ : \ \left| \frac{1}{2^{d+1}\sqrt{d}} \cdot \mathbf{1}_d^\top(\mathbf{w} - \mathbf{w}_1^*) \right| \leq \varepsilon \right\}$, we may observe that the entire set of $\varepsilon$-optimal classifiers for $L_{\mathcal{D}_1}$ is contained in $S_1$. $\qquad\square$

**Lemma 11.** *Let* $\mathbf{w}_2^* = \frac{1}{d-1} \cdot \mathbf{1}_d$, *and let* $S_2 = \left\{ \mathbf{w} \ : \ \left| \frac{1}{2^{d+1}\sqrt{d}} \cdot \mathbf{1}_d^\top(\mathbf{w} - \mathbf{w}_2^*) \right| \leq \varepsilon \right\}$. *Then the entire set of $\varepsilon$-optimal classifiers for* $L_{\mathcal{D}_2}$ *is contained in* $S_2$.

*Proof.* The proof is nearly identical to that of Lemma 10. We can again rewrite $L_{\mathcal{D}_2}$ as

$$L_{\mathcal{D}_2}(\mathbf{w}) = \frac{1}{2^d\sqrt{d}} \sum_{\mathbf{x}\in\mathcal{X}_2} \left(\frac{1}{2} + \delta\right)(1 - \mathbf{w}^\top\mathbf{x} + |1 - \mathbf{w}^\top\mathbf{x}|) + \left(\frac{1}{2} - \delta\right)(1 + \mathbf{w}^\top\mathbf{x} + |1 + \mathbf{w}^\top\mathbf{x}|)$$

Note that

$$\partial L_{\mathcal{D}_2}(\mathbf{w}) = \frac{1}{2^d\sqrt{d}} \sum_{\mathbf{x}\in\mathcal{X}_2} \left(\frac{1}{2} + \delta\right)(-\mathbf{x} - \text{sign}(1 - \mathbf{w}^\top\mathbf{x})\cdot\mathbf{x}) + \left(\frac{1}{2} - \delta\right)(\mathbf{x} + \text{sign}(1 + \mathbf{w}^\top\mathbf{x})\cdot\mathbf{x})$$

$$= \frac{1}{2^d\sqrt{d}} \sum_{\mathbf{x}\in\mathcal{X}_2} -2\delta\mathbf{x} - \left(\frac{1}{2} + \delta\right)\text{sign}(1 - \mathbf{w}^\top\mathbf{x})\cdot\mathbf{x} + \left(\frac{1}{2} - \delta\right)\text{sign}(1 + \mathbf{w}^\top\mathbf{x})\cdot\mathbf{x}$$

$$= \frac{1}{\sqrt{d}}\left(-\frac{\delta}{2}\mathbf{1}_d - \frac{1}{2^d}\sum_{\mathbf{x}\in\mathcal{X}_2}\left(\frac{1}{2} + \delta\right)\text{sign}(1 - \mathbf{w}^\top\mathbf{x})\cdot\mathbf{x} + \left(\frac{1}{2} - \delta\right)\text{sign}(1 + \mathbf{w}^\top\mathbf{x})\cdot\mathbf{x}\right)$$

Let $\mathcal{X}^* = \{\mathbf{x} : \|\mathbf{x}\|_1 = d - 1, \mathbf{x} \in \mathcal{X}_2\}$, and let $\mathbf{w}_2^* = \frac{1}{d-1}\mathbf{1}_d$. Taken together, we see that

$$\partial L_{\mathcal{D}_2}(\mathbf{w}_2^*) = \frac{1}{\sqrt{d}}\left(-\frac{\delta}{2}\mathbf{1}_d - \frac{1}{2^d}\sum_{\mathbf{x}\in\mathcal{X}^*}\left(\frac{1}{2} + \delta\right)\text{sign}(1 - \frac{1}{d-1}\mathbf{1}_d^\top\mathbf{x})\cdot\mathbf{x}\right.$$

$$- \frac{1}{2^d}\sum_{\mathbf{x}\in\mathcal{X}_2\setminus\mathcal{X}^*}\left(\frac{1}{2} + \delta\right)\text{sign}(1 - \frac{1}{d-1}\mathbf{1}_d^\top\mathbf{x})\cdot\mathbf{x}$$

$$\left.+ \frac{1}{2^d}\sum_{\mathbf{x}\in\mathcal{X}_2}\left(\frac{1}{2} - \delta\right)\text{sign}(1 + \frac{1}{d-1}\mathbf{1}_d^\top\mathbf{x})\cdot\mathbf{x}\right)$$

$$= \frac{1}{\sqrt{d}}\left(-\frac{\delta}{2}\mathbf{1}_d - \frac{d}{2^d}\left(\frac{1}{2} + \delta\right)\text{sign}(0)\cdot\mathbf{1}_d - \left(\frac{1}{4} - \frac{d}{2^d}\right)\left(\frac{1}{2} + \delta\right)\cdot\mathbf{1}_d + \frac{1}{4}\left(\frac{1}{2} - \delta\right)\cdot\mathbf{1}_d\right)$$

$$= \frac{1}{\sqrt{d}}\left(-\delta + (1 - \text{sign}(0))\left(\frac{d(1 + 2\delta)}{2^{d+1}}\right)\right)\cdot\mathbf{1}_d$$

Since $\delta = \frac{1}{2^{d+1}}$, and letting $\text{sign}(0) = 1$, we have that

$$\mathbf{h}^- = -\frac{1}{2^{d+1}\sqrt{d}} \cdot \mathbf{1}_d \in \partial L_{\mathcal{D}_2}(\mathbf{w}_2^*).$$

Furthermore, when $\text{sign}(0) = -1$, we have

$$\mathbf{h}^+ = \frac{1}{\sqrt{d}}\left(\frac{2d - 1}{2^{d+1}} + \frac{d}{4^d}\right) \cdot \mathbf{1}_d \in \partial L_{\mathcal{D}_2}(\mathbf{w}_2^*).$$

Since $\mathbf{0}_d$ can be written as a convex combination of $\mathbf{h}^-$ and $\mathbf{h}^+$, we know that $\mathbf{w}_2^*$ is a minimizer of $L_{\mathcal{D}_2}(\mathbf{w})$. Since $d \geq 4$, we also know that, for all $\mathbf{y} \in \mathbf{dom} L_{\mathcal{D}_2}$,

$$L_{\mathcal{D}_2}(\mathbf{y}) - L_{\mathcal{D}_2}(\mathbf{w}_2^*) \geq \frac{1}{2^{d+1}\sqrt{d}} \cdot \mathbf{1}_d^\top(\mathbf{y} - \mathbf{w}_2^*)$$

and

$$L_{\mathcal{D}_2}(\mathbf{y}) - L_{\mathcal{D}_2}(\mathbf{w}_2^*) \geq -\frac{1}{2^{d+1}\sqrt{d}} \cdot \mathbf{1}_d^\top(\mathbf{y} - \mathbf{w}_2^*)$$

which taken together implies that

$$L_{\mathcal{D}_2}(\mathbf{y}) - L_{\mathcal{D}_2}(\mathbf{w}_2^*) \geq \left| \frac{1}{2^{d+1}\sqrt{d}} \cdot \mathbf{1}_d^\top(\mathbf{y} - \mathbf{w}_2^*) \right|.$$

Again, letting $S_2 = \left\{ \mathbf{w} \; : \; \left| \frac{1}{2^{d+1}\sqrt{d}} \cdot \mathbf{1}_d^\top(\mathbf{w} - \mathbf{w}_2^*) \right| \leq \varepsilon \right\}$, we see that the entire set of $\varepsilon$-optimal classifiers for $L_{\mathcal{D}_2}$ is contained in $S_2$. □

We can now prove Theorem 5.

*Proof of Theorem 5.* Having proved that $S_1$ and $S_2$ contain the $\varepsilon$-optimal classifiers for $L_{\mathcal{D}_1}$ and $L_{\mathcal{D}_2}$, respectively, we now aim to show that for $\varepsilon < \frac{1}{2^{d+2}d^{3/2}}$, these sets are disjoint. Observe that for any $\mathbf{w} \in S_1$,

$$\left| \frac{1}{2^{d+1}\sqrt{d}} \cdot \mathbf{1}_d^\top(\mathbf{w} - \mathbf{w}_1^*) \right| \leq \frac{1}{2^{d+2}d^{3/2}} \quad \Rightarrow \quad \frac{1}{2^{d+1}} \cdot \mathbf{1}_d^\top(\mathbf{w} - \mathbf{w}_1^*) \leq \frac{1}{2^{d+2}d} \quad \Rightarrow \quad \mathbf{1}_d^\top\mathbf{w} \leq 1 + \frac{1}{2d}.$$

On the other hand, for any $\mathbf{w} \in S_2$,

$$\left| \frac{1}{2^{d+1}\sqrt{d}} \cdot \mathbf{1}_d^\top(\mathbf{w} - \mathbf{w}_2^*) \right| \leq \frac{1}{2^{d+2}d^{3/2}} \quad \Rightarrow \quad -\frac{1}{2^{d+1}} \cdot \mathbf{1}_d^\top(\mathbf{w} - \mathbf{w}_2^*) \leq \frac{1}{2^{d+2}d}$$

$$\Rightarrow \quad \mathbf{1}_d^\top\mathbf{w} \geq \frac{d}{d-1} - \frac{1}{2d} = 1 + \frac{d+1}{2d(d-1)} > 1 + \frac{1}{2d},$$

and so we see that no $\mathbf{w}$ can exist in both $S_1$ and $S_2$, which means we may conclude that the sets are disjoint. □

### A.3 Proof of Corollary 2

*Proof.* Suppose $0 < \varepsilon < \frac{1}{64}$. Let $\mathcal{X}_1, \mathcal{X}_2$ be the same as in the previous proof, let $\mathcal{X}_1' = \frac{1}{\sqrt{2}}\mathcal{X}_1$ and $\mathcal{X}_2' = \frac{1}{\sqrt{2}}\mathcal{X}_2$, and let $\mathcal{D}_1'$ and $\mathcal{D}_2'$ be uniform distributions over $\mathcal{X}_1' \times \{\frac{1}{\sqrt{2}}\}$ and $\mathcal{X}_2' \times \{\frac{1}{\sqrt{2}}\}$, respectively. Again, any learner observing $k < 2$ attributes cannot distinguish between the distributions. Note that $L_{\mathcal{D}_1'}(\mathbf{w}) = \frac{1}{2}L_{\mathcal{D}_1}$ and $L_{\mathcal{D}_2'}(\mathbf{w}) = \frac{1}{2}L_{\mathcal{D}_2}$.

The set of $\varepsilon$-optimal regressors for $L_{\mathcal{D}_1'}$ is $S_1 = \{\mathbf{w} : |\mathbf{w}^\top\mathbf{1}_2 - 1| \leq 2\sqrt{2\varepsilon}\}$, and for $L_{\mathcal{D}_2'}$ the set is $S_2 = \{\mathbf{w} : \|\mathbf{w} - \mathbf{1}_2\|_2 \leq 2\sqrt{2\varepsilon}\}$. Thus, for $\varepsilon < \frac{1}{64}$, $S_1$ and $S_2$ are disjoint sets by the reasoning in the proof of Theorem 1, and so a learner can distinguish between the two sets, which is a contradiction. □

### A.4 Proof of Corollary 4

*Proof.* Let $\mathcal{X}_1$ and $\mathcal{X}_2$ be the same as described in the proof of Theorem 3, and let $\mathcal{X}_1' = \frac{1}{\sqrt{d}}\mathcal{X}_1$ and $\mathcal{X}_2' = \frac{1}{\sqrt{d}}\mathcal{X}_2$. Let $\mathcal{D}_1'$ and $\mathcal{D}_2'$ be uniform distributions over $\mathcal{X}_1' \times \{\frac{1}{\sqrt{d}}\}$ and $\mathcal{X}_2' \times \{\frac{1}{\sqrt{d}}\}$, respectively. Note that $L_{\mathcal{D}_1'}(\mathbf{w}) = \frac{1}{\sqrt{d}}L_{\mathcal{D}_1}(\mathbf{w})$ and $L_{\mathcal{D}_2'}(\mathbf{w}) = \frac{1}{\sqrt{d}}L_{\mathcal{D}_2}(\mathbf{w})$. Scaling the function does not change the minimizer, so most of the proof of Theorem 3 still holds. The main difference is that the subgradient set is scaled by $\frac{1}{\sqrt{d}}$, which leads to an additional $\frac{1}{\sqrt{d}}$ factor in the precision bound. □

# B  Analysis of Algorithm 1

To prove Theorem 8, we first need a few intermediate lemmas.

**Lemma 12.** *For all $t \in [m]$ it holds that $\mathbb{E}_t[\|\hat{\mathbf{g}}_t\|_2^2] \leq H^2$.*

*Proof.* We have $\mathbb{E}_t[\|\hat{\mathbf{g}}_t\|_2^2] = \mathbb{E}_t[\|\ell'(\mathbf{w}_t^\top \tilde{\mathbf{x}}_t - y_t) \cdot \tilde{\mathbf{x}}_t\|_2^2] \leq H^2 \mathbb{E}_t[\|\tilde{\mathbf{x}}_t\|_2^2] \leq H^2$ . □

**Lemma 13.** *Let $\ell_t(\mathbf{w}) := \ell(\mathbf{w}^\top \mathbf{x}_t - y_t)$, where $\ell : \mathbb{R} \mapsto \mathbb{R}$ is a convex, H-Lipschitz function defined over the entire real line, and let $\tilde{\ell}_t(\mathbf{w}) = \mathbb{E}_{\tilde{\mathbf{x}}_t}[\ell(\mathbf{w}^\top \tilde{\mathbf{x}}_t - y_t)]$. Then, for all $\mathbf{w}$,*

$$|\ell_t(\mathbf{w}) - \tilde{\ell}_t(\mathbf{w})| \leq HB\sqrt{1 - \frac{k}{d}}.$$

*Proof.* Note that $|\ell_t(\mathbf{w}) - \tilde{\ell}_t(\mathbf{w})| = |\mathbb{E}_{\tilde{\mathbf{x}}_t}[\ell(\mathbf{w}^\top \mathbf{x}_t - y_t) - \ell(\mathbf{w}^\top \tilde{\mathbf{x}}_t - y_t)]|$. By Jensen's inequality we have

$$\left|\mathbb{E}_{\tilde{\mathbf{x}}_t}[\ell(\mathbf{w}^\top \mathbf{x}_t - y_t) - \ell(\mathbf{w}^\top \tilde{\mathbf{x}}_t - y_t)]\right| \leq \mathbb{E}_{\tilde{\mathbf{x}}_t}\left[|\ell(\mathbf{w}^\top \mathbf{x}_t - y_t) - \ell(\mathbf{w}^\top \tilde{\mathbf{x}}_t - y_t)|\right].$$

Observe that, since $\|\mathbf{x}_t\|_2 \leq 1$ and for any coordinate $i \in [d]$, $\mathbb{E}_{\tilde{\mathbf{x}}_t}[\tilde{\mathbf{x}}_t[i]] = \frac{k}{d}\mathbf{x}_t[i]$,

$$\mathbb{E}_{\tilde{\mathbf{x}}_t}[\|\mathbf{x}_t - \tilde{\mathbf{x}}_t\|_2] \leq \left(\mathbb{E}_{\tilde{\mathbf{x}}_t}\left[\sum_{i=1}^d (\mathbf{x}_t[i] - \tilde{\mathbf{x}}[i])^2\right]\right)^{\frac{1}{2}} = \left(\sum_{i=1}^d \mathbf{x}_t[i]^2 - \frac{2k}{d}\mathbf{x}_t[i]^2 + \mathbb{E}_{\tilde{\mathbf{x}}_t}\left[\tilde{\mathbf{x}}[i]^2\right]\right)^{\frac{1}{2}}$$

$$\leq \left(\sum_{i=1}^d \left(1 - \frac{k}{d}\right)\mathbf{x}_t[i]^2\right)^{\frac{1}{2}}$$

$$\leq \sqrt{1 - \frac{k}{d}}.$$

Along with the fact that $\ell$ is $H$-Lipschitz, this gives us

$$\mathbb{E}_{\tilde{\mathbf{x}}_t}\left[|\ell(\mathbf{w}^\top \mathbf{x}_t - y_t) - \ell(\mathbf{w}^\top \tilde{\mathbf{x}}_t - y_t)|\right] \leq H \cdot \mathbb{E}_{\tilde{\mathbf{x}}_t}\left[\|\mathbf{w}^\top(\mathbf{x}_t - \tilde{\mathbf{x}}_t)\|_2\right]$$

$$\leq H \cdot \|\mathbf{w}\|_2 \mathbb{E}_{\tilde{\mathbf{x}}_t}\left[\|\mathbf{x}_t - \tilde{\mathbf{x}}_t\|_2\right]$$

$$\leq HB\sqrt{1 - \frac{k}{d}} . \qquad □$$

**Lemma 14.** *Let $\ell_t(\mathbf{w}) := \ell(y_t \cdot \mathbf{w}^\top \mathbf{x}_t)$, where $\ell : \mathbb{R} \mapsto \mathbb{R}$ is a convex, H-Lipschitz function defined over the entire real line, and let $\tilde{\ell}_t(\mathbf{w}) = \mathbb{E}_{\tilde{\mathbf{x}}_t}[\ell(y_t \cdot \mathbf{w}^\top \tilde{\mathbf{x}}_t)]$. Then, for all $\mathbf{w}$,*

$$|\ell_t(\mathbf{w}) - \tilde{\ell}_t(\mathbf{w})| \leq HB^2\sqrt{1 - \frac{k}{d}}.$$

*Proof.* Note that $|\ell_t(\mathbf{w}) - \tilde{\ell}_t(\mathbf{w})| = |\mathbb{E}_{\tilde{\mathbf{x}}_t}[\ell(y_t \cdot \mathbf{w}^\top \mathbf{x}_t) - \ell(y_t \cdot \mathbf{w}^\top \tilde{\mathbf{x}}_t)]|$. By Jensen's inequality we have

$$\left|\mathbb{E}_{\tilde{\mathbf{x}}_t}[\ell(y_t \cdot \mathbf{w}^\top \mathbf{x}_t) - \ell(y_t \cdot \mathbf{w}^\top \tilde{\mathbf{x}}_t)]\right| \leq \mathbb{E}_{\tilde{\mathbf{x}}_t}\left[|\ell(y_t \cdot \mathbf{w}^\top \mathbf{x}_t) - \ell(y_t \cdot \mathbf{w}^\top \tilde{\mathbf{x}}_t)|\right].$$

Using the fact that $\mathbb{E}_{\tilde{\mathbf{x}}_t}[\|\mathbf{x}_t - \tilde{\mathbf{x}}_t\|_2] \leq \sqrt{1 - \frac{k}{d}}$ from the proof of Lemma 13, with the fact that $\ell$ is $H$-Lipschitz, we have

$$\mathbb{E}_{\tilde{\mathbf{x}}_t}\left[|\ell(y_t \cdot \mathbf{w}^\top \mathbf{x}_t) - \ell(y_t \cdot \mathbf{w}^\top \tilde{\mathbf{x}}_t)|\right] \leq H \cdot \mathbb{E}_{\tilde{\mathbf{x}}_t}\left[\|y_t \cdot \mathbf{w}^\top(\mathbf{x}_t - \tilde{\mathbf{x}}_t)\|_2\right]$$

$$\leq HB \cdot \|\mathbf{w}\|_2 \mathbb{E}_{\tilde{\mathbf{x}}_t}\left[\|\mathbf{x}_t - \tilde{\mathbf{x}}_t\|_2\right]$$

$$\leq HB^2\sqrt{1 - \frac{k}{d}} . \qquad □$$

Having proven the lemmas, we proceed to prove Theorem 8.

*Proof.* For the regression case, let $\tilde{\ell}_t(\mathbf{w}) = \mathbb{E}_{\tilde{\mathbf{x}}_t}[\ell(\mathbf{w}^\top \tilde{\mathbf{x}}_t - y_t)]$, where the expectation is over the random choice of indices in the construction of $\tilde{\mathbf{x}}_t$. For the classification case, let $\tilde{\ell}_t(\mathbf{w}) = \mathbb{E}_{\tilde{\mathbf{x}}_t}[\ell(y_t \cdot \mathbf{w}^\top \tilde{\mathbf{x}}_t)]$. Note that in either case, $\nabla \tilde{\ell}_t(\mathbf{w}_t) = \mathbb{E}_{\tilde{\mathbf{x}}_t}[\hat{\phi}_t \cdot \tilde{\mathbf{x}}_t] = \mathbb{E}_{\tilde{\mathbf{x}}_t}[\hat{\mathbf{g}}_t]$. From the standard OGD analysis [13] applied to $\hat{\mathbf{g}}_1, \ldots, \hat{\mathbf{g}}_m$, we see that

$$\sum_{t=1}^{m} \hat{\mathbf{g}}_t^\top (\mathbf{w}_t - \mathbf{w}^*) \leq \frac{2B^2}{\eta} + \frac{\eta}{2} \sum_{t=1}^{m} \|\hat{\mathbf{g}}_t\|_2^2. \tag{1}$$

Let $\mathbb{E}_t[\cdot]$ be the conditional expectation with respect to the randomness up to time $t$, and let $G^2 := \max_t \mathbb{E}_t[\|\hat{\mathbf{g}}_t\|_2^2]$. After taking the expectation of Eq. (1) with respect to the randomness in the algorithm, we have

$$\mathbb{E}\left[ \sum_{t=1}^{m} \nabla \tilde{\ell}_t(\mathbf{w}_t)^\top (\mathbf{w}_t - \mathbf{w}^*) \right] \leq \frac{2B^2}{\eta} + \frac{\eta}{2} G^2 m.$$

Since $\tilde{\ell}_t(\mathbf{w})$ is convex, we have $\tilde{\ell}_t(\mathbf{w}_t) - \tilde{\ell}_t(\mathbf{w}^*) \leq \nabla \tilde{\ell}_t(\mathbf{w})^\top (\mathbf{w}_t - \mathbf{w}^*)$. After choosing $\eta = \frac{2B}{G\sqrt{m}}$ and rearranging, we have

$$\mathbb{E}\left[ \sum_{t=1}^{m} \tilde{\ell}_t(\mathbf{w}_t) \right] \leq \sum_{t=1}^{m} \tilde{\ell}_t(\mathbf{w}^*) + 2BG\sqrt{m}.$$

Note that by Lemma 12, $G \leq H$. After dividing through by $m$ we have

$$\mathbb{E}\left[ \frac{1}{m} \sum_{t=1}^{m} \tilde{\ell}_t(\mathbf{w}_t) \right] \leq \frac{1}{m} \sum_{t=1}^{m} \tilde{\ell}_t(\mathbf{w}^*) + \frac{2HB}{\sqrt{m}}.$$

Let $\tilde{B} = \max\{B, 1\}$. By Lemmas 13 and 14, this implies that

$$\mathbb{E}\left[ \frac{1}{m} \sum_{t=1}^{m} \ell_t(\mathbf{w}_t) \right] \leq \frac{1}{m} \sum_{t=1}^{m} \ell_t(\mathbf{w}^*) + \frac{2HB}{\sqrt{m}} + 2H\tilde{B}^2 \sqrt{1 - \frac{k}{d}}.$$

Taking the expectation with respect to the random choice of the data set, we have

$$\mathbb{E}\left[ \frac{1}{m} \sum_{t=1}^{m} L_{\mathcal{D}}(\mathbf{w}_t) \right] \leq L_{\mathcal{D}}(\mathbf{w}^*) + \frac{2HB}{\sqrt{m}} + 2H\tilde{B}^2 \sqrt{1 - \frac{k}{d}}.$$

After applying Jensen's inequality, we have

$$\mathbb{E}[L_{\mathcal{D}}(\hat{\mathbf{w}})] \leq L_{\mathcal{D}}(\mathbf{w}^*) + \frac{2HB}{\sqrt{m}} + 2H\tilde{B}^2 \sqrt{1 - \frac{k}{d}}. \qquad \square$$