[Reviews · NeurIPS 2016]

Reviewer 1

Summary

The paper shows lower bounds for the problem of learning with missing attributes. In this problem, one is given examples (x,y) sampled from some distribution on R^d \times R. The goal is to find a linear classifier that minimizes a certain lose (e.g., the square loss or the hinge loss). The twist in this framework is that the learner cannot see all features, but rather, for each example, it can choose to see at most k of the features. This setting has been suggested by Ben David and Dichterman in 1998, and since then algorithms were found for the square loss, even in the case that k=2. On the other hand, for other loss functions, there are no known algorithms for fixed k. This paper shows that indeed such algorithms do not exist. Namely, it is shown that for the square loss, learning is impossible for k=1 (so that k-2 is tight). For the absolute and the hinge loss, it is shown that learning is impossible for every fixed k. The proof goes by exhibiting two distributions that are indistinguishable given this limited access to examples. The proof is rather simple in the case of the square loss, but more delicate in the case of the absolute and hinge losses.

Qualitative Assessment

The paper makes a solid contribution to the problem at hand. Hence, I'd recommend acceptance.

Confidence in this Review

2-Confident (read it all; understood it all reasonably well)


Reviewer 2

Summary

The manuscript provides lower bounds for regression (and classification) in the limited attributes observation model. In particular, the lower bounds tighten what is previously known for the squared loss in this setting, and novel lower bounds for absolute loss and hinge loss (in classification) are shown. An algorithm that achieves loss bound up to a certain precision limit is also provided. The results are sufficiently novel and the paper is well-written.

Qualitative Assessment

The authors provide lower bounds for regression and classification in the limited attributes observation model. Here, the learning algorithm has access to a subset of attributes, and we want to control the excess loss (over the bayes optimal). The manuscript provides lower bounds in this setting that improve over known bounds, and resolve some open questions in this setting. The manuscript makes interesting contributions: the provided lower bounds tighten what is previously known for the squared loss in this setting, and for absolute loss and hinge loss (in classification), novel lower bounds are derived. Furthermore, an algorithm that achieves loss bound up to a certain precision limit is also provided; this matches lower-bound up-to a polynomial factor in dimensionality (in case of regression). The results and proof techniques are sufficiently novel and the paper is well-written. I recommend acceptance. The results in classification setting, albeit, appear less interesting (there is an exponential gap between the lower bound and the guaranteed precision limits of the proposed algorithm for hinge loss). The authors do not comment on Theorem 5 after it is briefly stated in Section 2.1. In particular, the nature of the result in Theorem 5 is different from that in Theorems 1 and 3 (for regression), and seem weaker (existence of some epsilon vs actually quantifying the limits of precision). I would like the authors to comment on this aspect, and perhaps an exposition in the main paper as to the difficulty of stating a stronger result (if that is the case) would be useful. (I've read the author response.)

Confidence in this Review

2-Confident (read it all; understood it all reasonably well)


Reviewer 3

Summary

The paper presents some lower bounds on the ability to learn when only part of the attributes of each sample is visible to the learner. The scenario that the paper investigated had bee explored previously and the paper explain why the results presented does not contradict known results. Finally the paper present a general generic algorithm and show an upper bound on the error achieved.

Qualitative Assessment

Solid paper, provide important results in a clear and precise way. Good review of prior knowledge in this field. I do believe however that the authors can better explain at least for the first lower bound ( general cost function) why do things go wrong. It seems to me that in most cases learning is achievable, however not in all cases. Better describing why the learning process fails can give researches much needed insight which in my view is much more valuable then the precise mathematical lower limit.

Confidence in this Review

2-Confident (read it all; understood it all reasonably well)


Reviewer 4

Summary

This paper derived the lower bound of attributes needed to achieve some precision levels, when the regression/classification models with different loss functions can only access limited attributes.

Qualitative Assessment

The paper is well-written, and the theoretical proofs in the paper and appendix look solid to me. However, it would be more convincing if the authors can provide supportive experimental results. And I feel some description could be added in the appendix about the general ideas of proofs, e.g. why the theorems can be proved from fact that sets S_1 and S_2 are disjoint. In the proofs, the elements in vector x are either 0 or 1, I am curious about whether the results change under assumption that x belongs to [0, 1]^d.

Confidence in this Review

1-Less confident (might not have understood significant parts)


Reviewer 5

Summary

In this paper, the authors present the first lower bounds giving a limit on the precision attainable by any algorithm for several variants of regression, as well as for classification with the hinge loss, in the condition of limited attribution observation. The demonstration of these theorems is integrated.

Qualitative Assessment

But the authors did not give any experiments to verify these conclusions, and only prove them reasonable in theory.

Confidence in this Review

2-Confident (read it all; understood it all reasonably well)